# Profiling Glioblastoma Cases with an Expression of DCX, OLIG2 and NES

**DOI:** 10.3390/ijms222413217

**Published:** 2021-12-08

**Authors:** Adrian Odrzywolski, Bożena Jarosz, Michał Kiełbus, Ilona Telejko, Dominik Ziemianek, Sebastian Knaga, Radosław Rola

**Affiliations:** 1Department of Biochemistry and Molecular Biology, Medical University of Lublin, 20-093 Lublin, Poland; adrian.odrzywolski@umlub.pl (A.O.); michal.kielbus@umlub.pl (M.K.); ilona.telejko@umlub.pl (I.T.); 2Laboratory for Cytogenetics and Genome Research, Department of Human Genetics, KU Leuven, B-3000 Leuven, Belgium; 3Department of Neurosurgery and Pediatric Neurosurgery, Medical University of Lublin, 20-090 Lublin, Poland; bozena.jarosz@umlub.pl (B.J.); ziemianek.dominic@gmail.com (D.Z.); 4Institute of Biological Bases of Animal Production, University of Life Sciences, 20-950 Lublin, Poland; sebastian.knaga@up.lublin.pl

**Keywords:** Glioblastoma, Doublecortin, DCX, OLIG2, NES, single cell RNA-seq, immunohistochemistry, immunofluorescence, brain tumours

## Abstract

Glioblastoma (GBM) remains the leading cause of cancer-related deaths with the lowest five-year survival rates among all of the human cancers. Multiple factors contribute to its poor outcome, including intratumor heterogeneity, along with migratory and invasive capacities of tumour cells. Over the last several years Doublecortin (DCX) has been one of the debatable factors influencing GBM cells’ migration. To resolve DCX’s ambiguous role in GBM cells’ migration, we set to analyse the expression patterns of DCX along with Nestin (NES) and Oligodendrocyte lineage transcription factor 2 (OLIG2) in 17 cases of GBM, using immunohistochemistry, followed by an analysis of single-cell RNA-seq data. Our results showed that only a small subset of DCX positive (DCX^+^) cells was present in the tumour. Moreover, no particular pattern emerged when analysing DCX^+^ cells relative position to the tumour margin. By looking into single-cell RNA-seq data, the majority of DCX^+^ cells were classified as non-cancerous, with a small subset of cells that could be regarded as glioma stem cells. In conclusion, our findings support the notion that glioma cells express DCX; however, there is no clear evidence to prove that DCX participates in GBM cell migration.

## 1. Introduction

Even though malignant brain tumours account only for a small percentage of all adult cancers, they lead to an extensive amount of cancer-related deaths [1]. Moreover, the five-year survival rates are among the lowest for all human cancers [2], regardless of treatment modality [3]. This remarkable resistance results mostly from tumour heterogeneity and its high propensity for malignant progression. One of the vital pathophysiologic features contributing to this dismal prognosis is their strong migrational capacity [4] for significant dispersal beyond the macroscopic tumour borders [5]. Interestingly, a similar migratory ability is one of the principal features of neuronal progenitor cells (NPC) during CNS development [6,7]. Given that, it is worth noticing that data generated by the Cancer Genome Atlas Research Network proved that one of the main gene profiles of GBM, a proneural profile, involves DCX expression [8].

DCX itself, when it is mutated, is responsible for an X-linked form of lissencephaly, affecting the organisation of neocortical layering in the cerebral cortex [9]. Subsequent studies have shown that DCX directly binds to microtubules, thereby regulating their polymerisation and stabilisation [10]. This process is crucial for a multipolar mode of neuroblast migration [11], a transient stage in neuronal progenitor migration where migrating cells search for environmental signals that will determine their mode of migration [12]. DCX expression has been restricted to migrating neuroblasts in developing and adult animals [12]. However, Daou et al. [13] proved that DCX expression might be found in various neuroepithelial origin tumours. 

Interestingly, DCX was highly expressed in both high-grade and low-grade invasive tumours. Moreover, invasive tumours have been shown to express higher levels of DCX when compared to circumscribed tumours; no expression in normal brain tissue surrounding the tumour was found. A more recent follow-up study evaluated the sensitivity and specificity of DCX immunostaining to detect infiltrating glioma cells [14]. It confirmed that DCX is explicitly expressed in infiltrating gliomas but not in reactive astrocytes. Santra et al. presented a different hypothesis [15]. Their data indicate that DCX mRNA transcripts were not detected in primary glioma cells, while DCX expressing cells were revealed in tumour penumbra. DCX positive cells within glioma tumours, in their opinion, were either infiltrating neuroblasts or pre-existing neuronal cells.

According to Verhaak et al. [8], one of the proneural subtype’s signature genes is Oligodendrocyte lineage transcription factor 2 (OLIG2), a family member of basic helix–loop–helix transcription factors. It plays a crucial role in the early stages of brain development in oligodendrocyte precursor cells and neural progenitor cells by enhancing mitosis and limiting cell differentiation [16]. In gliomas, OLIG2 was expressed by glioma stem cells [17]. On top of that, cells expressing Rai (ShcC/N-Shc), a member of the family of Shc-like adaptor proteins, are involved in non-neoplastic cell migration, co-expressed DCX, and OLIG2 [18]. Accordingly, whenever DCX and OLIG2 is observed in migrating cells, it would imply that these cells have stem-cell capabilities. 

On the other hand, Bott et al. [19] recently found that DCX function in complex with nestin (NES) is a marker of neural progenitors. Although initially detected in neuronal stem cells, its presence in multiple other tissues (including gliomas) has recently been described. NES is a type VI intermediate filament. It plays a role in several key aspects of primary cell functioning: self-renewal, proliferation, survival, differentiation, and migration. With regard to the later, Bott et al. established that cdk5/p35 selectively phosphorises DCX due to the DCX-NES complex’s presence. That could directly influence the growth cone during migration.

Based on the observations mentioned above, we contend that the role of DCX expression in glioma cells’ migration is still a matter of debate. Therefore, in order to further clarify this role, we decided to conduct an experiment focusing on the distribution of glioma cells, both inside and in the tumour’s margin. Simultaneously, in order to elucidate the possible role of DCX in the context of the proneuronal subtype pathogenesis with its potential interactions with microtubules and intermediate filaments, we have marked OLIG2 and NES.

## 2. Results

Our heuristic approach to quantify the number of cells with each of the antigens allowed us to obtain data that can be compared with the results from single-cell RNA-seq. The fundamental component of this method is the determination of cell nuclei in the examined ROI. Cellular nuclei (Figure 1A) act as seeds to assess adjacent fluorescent signals: OLIG2 was expected to overlap with nuclei (Figure 1B,F,H), whereas NES (Figure 1C,E,H) and DCX (Figure 1D,G,H) signal should tightly adhere to it.

As expected, cell counts were significantly different in various parts of the tumour. More cells were tumour-adjacent to the margin or in non-specific tumour sites than in the margin (*p* = 1 × 10^−4^ and *p* = 1.9 × 10^−7^, respectively)—Figure 2H. No significant cell count change was detected while comparing tumours adjacent to margin and non-specific tumour sites (*p* = 0.91).

The DCX^+^ cells were only a small subpopulation of cells in total, although they were present in all types of tissues—Figure 2A. The median frequency of DCX^+^ cells was 3.1% in margin tissue and 1.1% in adjacent tumour tissue, without significant difference (*p* = 0.82). The majority of ROIs did not have a context of margin tissue. Thus, it was impossible to assess the exact placement within the tumour. They were grouped and called other/non-specific tumour tissue. A batch comparison of DCX^+^ cells in other tumour tissue versus tumour-adjacent to margin did not show statistical importance (*p* = 0.24).

Contrary to the frequency of DCX^+^ cells, NES^+^ cells were much more abundant in studied cases (Figure 2B). The median frequency was 17.7% in margin tissue, 20.6% in adjacent tumour tissue, and 25.4% in the rest of ROIs tumours, yet without significant difference (the lowest *p* = 0.78 in the batch comparison, and *p* = 0.30 when accounting pairs of margins vs. adjacent tumour).

OLIG2^+^ cells were more frequent than DCX^+^ cells (Figure 2C). Non-specific tumour sites and tumours close to the margin had a similar median (8.2% and 5.1%, respectively). The median of the tumour margin, on the other hand, was 19.8%. Pairwise comparison of the margin and the tumour-adjacent to the margin shows a significant change in OLIG2^+^ cells frequency (*p* = 4.9 × 10^−4^).

No significant changes in frequency in cells expressing more than one of the analysed markers were detected. The majority of measured frequencies were next to 0% (Figure 2D–G).

Furthermore, we tested if subsequent tumour resections impacted cell count with specific staining (Figure 3A–G). We have focused on the first three subsequent resections, as they have a representative number of cases. There were no significant trends in all stainings (DCX^+^: *p*-value = 0.90, OLIG2^+^: 0.72, NES^+^: *p*-value = 0.89; df = 2).

Then, we tested if there are any correlations between the frequency of cells with different antigens and sites (Appendix A). The frequency of cells expressing either NES or OLIG2 had a strong positive correlation between margins and tumour sites adjacent to margins. Although cells expressing DCX lacked this correlation, there was a strong negative correlation between the frequency of DCX^+^ cells in the margin and the frequency of NES^+^ cells both in tumour and margin sites.

Finally, we did not observe any change in overall survival regarding tested antigens (Appendix A).

To better understand the function of DCX, we opted for the single-cell approach, which allowed annotating the cells as “cancer” and “normal” according to the number of GBM specific copy number variations (CNV) prediction (Appendix A) based on transcriptomics of single cells (Figure 4A). To avoid false-positive identification of malignant cells, we considered only the canonical aberrations for GBM, which are a duplication of chromosome 7 or chromosome 10 loss. The cells defined as “cancer” and “normal” were 39.77% and 60.23%. The cells identified as ‘normal’ were clustered together, mainly in the two big clusters—one of them was specific for the cells derived from the foetal origin, whereas the second big cluster contained the non-cancer brain cells (Figure 4B–D). The ‘cancer’ cells, which were defined as GBM specific CNVs containing, had been clustered into the number of overlapping and sample-specific clusters, which shows their high heterogeneity (Figure 4A–C).

Based on the single-cell transcriptomics, we also annotated the cluster cells to define the analysed cells’ phenotype (Figure 5A–C). The complete list of cluster-specific genes are attached to the Appendix A chapter as a CSV file, whereas the distribution of the cells among the clusters by their “cancer” and “normal” status are shown in Appendix A. These annotations were used to predict the role of DCX, OLIG2, and NES in tumour biology. We visualised the expressions of these genes that were shown among defined cell clusters in Figure 6. The cells defined as “normal” showed the expression of specific markers for neurons, astrocytes, dendritic cells, macrophages, plasmacytoid dendritic cells, circulating foetal cells, undifferentiated cells, and oligodendrocytes. These cells that had been labelled as “cancer” expressed the markers specific for circulating foetal cells, basophils, astrocytes, dendritic cells, neurons, proliferative cells, undifferentiated cells, and oligodendrocytes. The cells’ distribution by their “cancer” or “normal” status is shown in Appendix A.

OLIG2 level in the cells annotated as “cancer” was found only in this cluster of cells which also expressed the markers specific for astrocytes, neurons, and oligodendrocytes; however, it also was found in the cells with undifferentiated and proliferative properties. NES was the most specific for the cells annotated as ‘cancer’, for which basophil markers are characteristic. DCX in the “cancer” identified cells was expressed in these clusters, identified according to the transcriptomics signatures specific for oligodendrocytes, undifferentiated, and proliferative cells (Figure 6A). In normal cells, OLIG2 was found mainly in the undifferentiated cells and in lower levels in clusters of proliferative cells and this group of cells with the gene signature that is specific for neurons, lymphocytes, and macrophages. NES expression was mostly seen in the clusters of cells that expressed the markers of neurons, lymphocytes, and macrophages and those that harboured the transcripts specific for proliferative and undifferentiated cells. DCX in normal cells was specific for neurons and astrocytes (Figure 6B).

We also evaluated the correlations between the expression of DCX, OLIG2, and NES genes at a single-cell level in the population of cells derived from the tumour tissue to show the potential relation of these genes. We found significant (*p* < 0.01) but rather weak positive correlations between transcript levels of these genes conducting Spearman’s rank test: DCX vs. OLIG2 (r = 0.2179733), NES vs. OLIG2 (r = 0.3019327), NES vs. DCX (r = 0.1535854). When we look closer at the populations of these cells that had been annotated as ‘normal’, simultaneous expression of OLIG2 and DCX was observed in the clusters of these cells that were expressing markers specific for highly proliferative or undifferentiated cells as well as in the cluster of cells expressing markers that are common for astrocytes, oligodendrocytes and neurons. On the other hand, in the population of cancerous cells, NES and DCX were simultaneously expressed by the cells that were positive mostly for markers of: proliferative, undifferentiated or astrocytic cells.

## 3. Discussion

DCX status in GBM has been a subject of debate since Rich [20] and Daou [13] reported the expression of DCX on mRNA and protein levels in 2005. While Rich et al. correlated the mRNA level of DCX with poor diagnosis, Daou et al. proved that DCX had more intense staining towards the margin of the tumour using immunostaining. Not all samples of GBM had the DCX expression, however. Interestingly, our results did not confirm most of those observations. Although DCX was detected in most samples, DCX^+^ cells were only a small subset of the GBM’s cell population, both in tumours and margins. Only in a few cases were there many DCX^+^ cells. Moreover, there was no shift in DCX^+^ cell frequency towards one of the sites. Although inconclusive, this may imply that DCX is not directly connected to GBM cell migration.

On the other hand, Santra et al. [15,21] reported DCX as a marker for a favourable patient outcome. They found that cells with DCX overexpression had lower invasion abilities, thus, the authors concluded that DCX positive cells in glioma either infiltrate neuroblasts or pre-existing neuronal cells. Importantly, our data partially supports this idea. Although most of the cells expressing DCX within the tumour were marked as non-cancerous, four subpopulations were marked as GBM origin. They followed expression patterns similar to oligodendrocytes, astrocytes, as well as proliferative and undifferentiated cells. This aligns with the idea that DCX is expressed by GSC.

In general, NES^+^ and OLIG2^+^ cells were more frequent than DCX^+^ ones, which is consistent with the bulk values reported in The Human Protein Atlas [22]. Interestingly, cells expressing NES or OLIG2 were present both in the margin and in tumours, while being highly correlated. The lack of a significant drop in NES^+^ cells is in contradiction with a study by Smith et al. [23]; this might be explained by a different definition of margin adopted in our study. Smith et al. defined margin as a region with 5-aminolevulinic acid (5ALA) fluorescence during surgery that is beyond the T1 enhancing region on magnetic resonance imaging (MRI) [23]. However, when considering regions by their relative position to tumour sites, both studies are consistent. We also found that OLIG2^+^ cells frequency differed among sites: more cells with OLIG2 expression were in margin than in tumour.

On the other hand, the frequency of NES^+^ cells on the margin and within the tumour were reversely correlated with DCX^+^ cells in the margin. A negative correlation seems not to support the existence of the DCX-NES complex in the tumour. It is worth keeping in mind that there is a gross difference between the frequency of DCX- and NES-positive cells, and sample size does not compensate for that. When we look closer at the populations of these cells that had been annotated as ‘normal’, simultaneous expression of OLIG2 and DCX was observed in the clusters of these cells that were expressing markers specific for highly proliferative or undifferentiated cells as well as in the cluster of cells expressing markers that are common for astrocytes, oligodendrocytes and neurons. On the other hand, in the population of cancerous cells, NES and DCX were simultaneously expressed by the cells that were positive mostly for markers of: proliferative, undifferentiated or astrocytic cells.

Another interesting finding was that subsequent resections did not significantly influence assessed markers’ frequency. In fact, all three of them seemed to be associated with some form of undifferentiated cells. It was reported previously that glioma stem cells have higher invasion capabilities [24], suggesting that stemness might be a phenotypic response to changing the tumour’s microenvironment [25]. Our data also support the notion that there is a set proportion of cells expressing DCX/NES/OLIG2, regardless of treatment, guided by tumour plasticity.

The GBM cells revealed high heterogeneity, which we saw as clustering into several different groups of cells derived by transcriptomics. Others also reported a similar relationship [26], which confirms the value of our analytical approach. However, annotating the cells to “cancer” or “normal” subpopulations using our quite simple approach may not be entirely accurate as we might omit other possible genetic aberrations than those used canonically for GMB. Nonetheless, our approach seems to be valuable, for it limits the false positive detection of malignant cells. Moreover, the algorithms used in our investigation allowed us to properly annotate the neurons and the immune cells (i.e., macrophages, dendritic and T cells) as non-cancer cells. However, we also identified a distinct cluster of cells that we recognised as “cancer” because of genetic aberrations parallel to CD63 basophil marker expression, while not being positive for other canonical basophil markers (e.g., CD123). That suggests that these cells should not be classified as immune cells [27].

On the technical side, our primary consideration was to distinguish individual cells. Fluorescent signals from cell nuclei and OLIG2 were easy to partition between cells, as in the vast majority of cases, it was single point luminescence. On the other hand, DCX fluorescence was more challenging to evaluate because of the branched structure of a cell’s cytoskeleton. Finally, the hardest to assess was NES: not only were the signal figures branched, but NES+ cells were also more prevalent, with a tendency to be clumped.

In conclusion, our findings support the notion that DCX is indeed expressed by glioma cells, but there is no clear evidence to prove that it may participate in GBM cell migration. Other GSC markers: NES and OLIG2 are in much higher abundance and are present both in tumours and their margin.

## 4. Materials and Methods

### 4.1. Samples Collection and Selection of Regions of Interest (ROIs)

Paraffin-embedded samples were obtained from the Department of Neurosurgery and Paediatric Neurosurgery archive, Medical University of Lublin (Poland). We selected only those samples subjected to at least two resections, and at least one sample from each patient was diagnosed with GBM. Moreover, we have focused on case studies since 2011. For each of the samples, a trained neuropathologist performed second-hand diagnosis, along with marking ROIs. The goal was to mark regions either with representative tumour tissue or with the border between tumour and margin tissue. In total, we collected and assessed 17 cases over 46 paraffin-embedded tissues, marking 60 ROIs.

Additionally, all ROIs were classified using two types of categories: tissue origin and surgery sequence number. Tissue origin contains four subcategories: GBM cases (1) when a tumour was adjacent to margin, (2) tissue adjacent to margin, (3) tumour-only when tumour ROI was far from margin/margin was not detected, and (4) tumour-only non-GBM, with ROIs containing a lower-grade astrocytoma. The second category showed surgery sequence number corresponding to samples taken during the first, second or third surgery.

### 4.2. Multiplex Immunofluorescence Staining

Paraffin-embedded tissues were cut into three µm sections and placed on glass slides (Thermo Fisher Scientific, Waltham, MA, USA: 10149870). Slides were baked overnight at 60 °C. The next day, slides were deparaffinised and hydrated in a series of xylene and ethyl alcohol. Antigens were retrieved by microwave-HIER in a citrate buffer (pH 6.0) over 20 min, followed by another 20 min in RT to cool down. Endogenous peroxide activity was quenched by a mixture of 1% H2O2 and 1% sodium azide for 20 min. All non-specific binding sites were blocked by incubation with blocking solution: PBS (Merck, Darmstadt, Germany: P4417-50TAB) with 1% BSA (Jackson ImmunoResearch, Cambridgeshire, UK: 001-000-161), 5% NDS (Cambridgeshire, UK: 017-000-121) and 1% Triton X-100, for 1 h. OLIG2, NES, and DCX were marked sequentially. Firstly, sections were incubated with appropriate primary antibodies (anti-NES: 1:100, Novus Biologicals, Centennial, CO, USA, MAB1259; anti-OLIG2: 1:100, AF2418; anti-DCX: 1:100, Cambridge, United Kingdom, ab18723; diluted in PBS with 1% BSA, 1% Triton X-100). After washing, corresponding secondary antibodies conjugated with HRP were used (anti-Mouse–Rabbit–Goat: 1:500, Cambridgeshire, UK, 715-036-150, 711-036-152, 705-036-147; diluted in PBS with 1% BSA, 1% Triton X-100). The visualisation was carried out using the Tyramide Signal Amplification (Thermo Fisher Scientific, Waltham, MA, USA: B40953, B40958, B40955) setup. In between fluorescent stainings, HRP activity was quenched by a mixture of 1% H2O2 and 1% sodium azide for 20 min to not interfere with subsequent staining. Finally, nuclei were detected with Hoechst-33342 (Thermo Fisher Scientific, Waltham, MA, USA: H3570), and slides were mounted (Thermo Fisher Scientific, Waltham, MA, USA: P36982). During incubations, slides were covered with a hand-cut parafilm piece, and all washing between steps was conducted with PBS 3 × 5 min. Incubations with primary and secondary antibodies lasted 1 h each.

### 4.3. Image Acquisition

All images were captured using a Nikon Ti Confocal microscope, using four lasers for fluorescence: 405, 488, 563, and 647 nm. The NISelements (ver 3.22.08, Melville, NY, USA) software was used to set up analyses. The corresponding primary antibody’s negative control was used to control non-specific staining for each batch of analysed slides. In this regard, specific laser intensity and gain parameters were chosen to remove any signal corresponding to 488, 563 and 647 nm. In addition, epifluorescence site conformation was done before confocal imaging to ensure the highest quality of the images.

### 4.4. Image Analysis

If possible, three different random square spots with a side of 600 pixels from each tissue image were selected to perform further analysis. In total, 152 random square spots were generated across 60 ROIs.

First, stack images were divided by channels corresponding to the wavelength used to detect each antigen: NES, OLIG2, DCX, and nuclei. Next, all features were marked, applying the following three heuristic assumptions:-Nucleus—any oval and coherent figure with a signal pattern corresponding to Hoechst-33342;-DCX/NES—nucleus with an adjacent signal corresponding to DCX or NES; in case of a signal adjacent to more than one nucleus, all were counted as positive;-OLIG2—nucleus overlapping with a signal corresponding to OLIG2.

A nuclei drove Voronoi’s diagram was created to find all positive cells for each of the fluorescent signals. Finally, DCX/NES/OLIG2 layers with marks were used to find and count all overlaps with cells in Voronoi’s diagram. The analysis was carried out in ImageJ (Fiji) [28].

### 4.5. Image Deposition and Sharing

We set up a local data share service based on Digital Slide Archive (DSA) [29]. All images in TIFF format were converted to pyramidal TIFF format using the ImageMagick tool [30] and uploaded to DSA along with all necessary metadata.

### 4.6. Single-Cell RNA-Seq and Data Processing

The data used here were generated and published previously by Couturier et al. [26]. It was further processed using the CellRanger pipeline. We included data for 12 CRC patients (originally named as OPK333B, OPK338B, OPK346B, OPK363, OPK364B, OPK368B, OPK389B, OPK390, OPK397, OPK402B, OPK407, OPK409) and three samples derived from the foetal brain (HFA567, HFA570, HFA571). First, the raw gene expression matrix was filtered and normalised using the Seurat R package. Then, the dataset was filtered according to the following criteria: cells with >1000 unique molecular identifier (UMI) counts; >500 genes and <5000 genes; and <10% of mitochondrial gene expression in UMI counts. The gene expression matrices from 15,000 randomly down-sampled cells were log-normalised to the total UMI counts per cell, scaled and finally clustered and visualised using t-SNE projection. The major cell types were characterised by comparing the canonical marker genes found in tissue-specific cell taxonomy reference database CellMatch and the differentially expressed genes for each cluster using the scCATCH automatic annotation algorithm [31].

To identify evidence for somatic large-scale chromosomal copy number alterations, we used inferCNV of the Trinity CTAT Project algorithm [32] in the reference with non-cancer, foetal brain cells. The approach used to annotate the cells as malignant (‘cancer’), was based on the presence of the most common canonical aberration for GBM (duplication of chromosome 7 or chromosome 10 loss), and it is line with the analytical strategy used previously by Couturier et al. [26].

### 4.7. Statistics

Data files from image analysis were imported and analysed in Rstudio, along with ggplot, corrplot, and dplyr packages. Pairwise comparisons using the Wilcoxon rank-sum test was used to check the cell count in different regions of samples. The Wilcoxon signed-rank test was used to find any consequential difference between margin and adjacent tumour tissue in terms of any staining. Mann–Whitney-Wilcoxon Test compared any other tumour tissue to the ones that were adjacent to the margin. The Kruskal–Wallis rank-sum test allowed to check any change in staining after subsequent tumour resections. Spearman’s rank correlation coefficient measured correlations. To check overall survival, Cox proportional-hazards models were used. In all tests, *p* < 0.05 was considered significant. All subsamples of the same image were averaged.

## Figures and Tables

**Figure 1 ijms-22-13217-f001:**
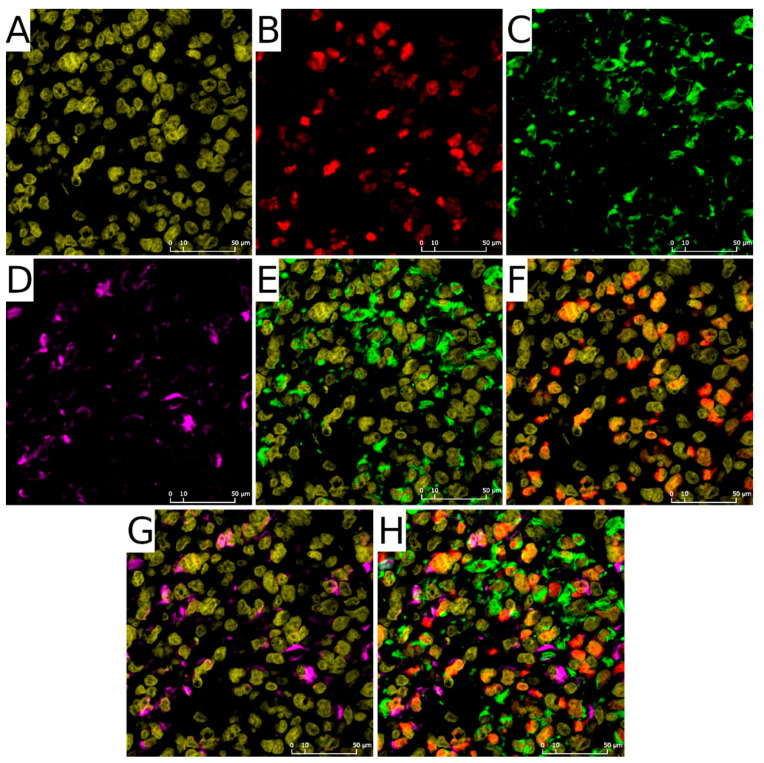
Staining patterns. (**A**)—Nuclei (yellow), (**B**)—OLIG2 signal (red), (**C**)—NES signal (green), (**D**)—DCX signal (magenta), (**E**)—NES signal with nuclei, (**F**)—OLIG2 signal with nuclei, (**G**)—DCX signal with nuclei (**H**)—composition of (**A**–**D**).

**Figure 2 ijms-22-13217-f002:**
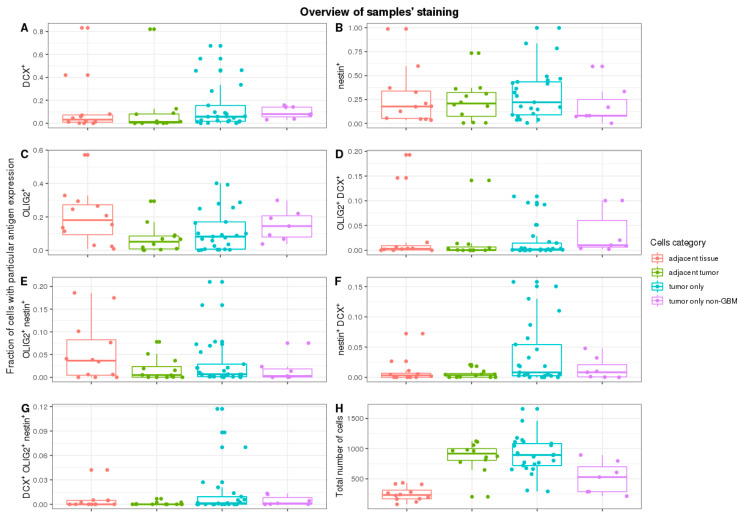
Overview of samples’ staining and staining overlap. Frequency of cells with different stainings with respect to tissues: (**A**)—DCX^+^ cells, (**B**)—NES^+^ cells, (**C**)—OLIG2^+^ cells. Frequency of cells with mixtures of all stainings with respect to tissues: (**D**)—OLIG2^+^DCX^+^ cells, (**E**)—OLIG2^+^NES^+^ cells, (**F**)—NES^+^DCX^+^ cells, (**G**)—DCX^+^OLIG2^+^NES^+^ cells. (**H**)—Total number of cells irrespective of antigen staining.

**Figure 3 ijms-22-13217-f003:**
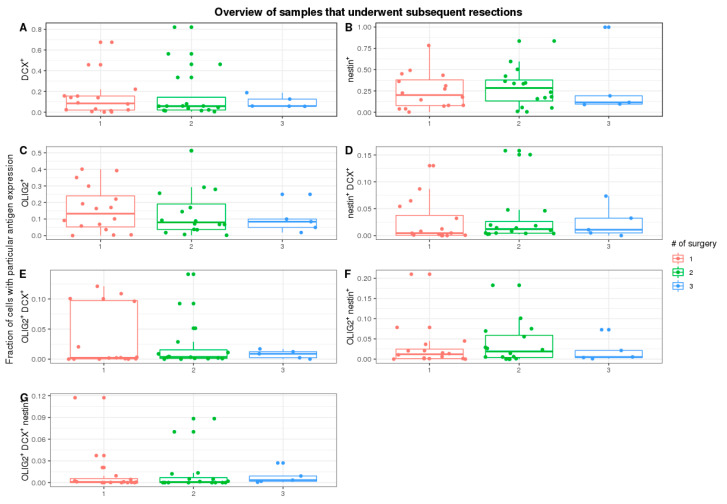
Overview of samples underwent subsequent resections. Frequency of cells with different stainings for tissues: (**A**)—DCX^+^ cells, (**B**)—NES^+^ cells, (**C**)—OLIG2^+^ cells. Frequency of cells with mixtures of all stainings for tissues: (**D**)—OLIG2^+^DCX^+^ cells, (**E**)—OLIG2^+^NES^+^ cells, (**F**)—NES^+^DCX^+^ cells, (**G**)—DCX^+^OLIG2^+^NES^+^ cells.

**Figure 4 ijms-22-13217-f004:**
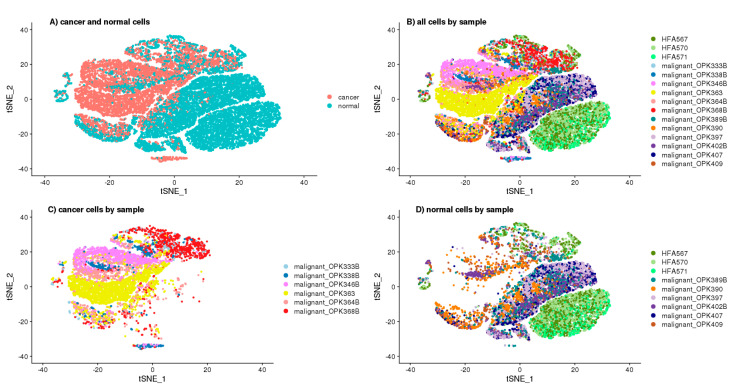
tSNE projection of single cells. (**A**)—Cancer and normal cells, (**B**)—All the cells coloured by sample name, (**C**)—“Cancer” cells only coloured by sample name, (**D**)—“Normal” cells only coloured by sample name.

**Figure 5 ijms-22-13217-f005:**
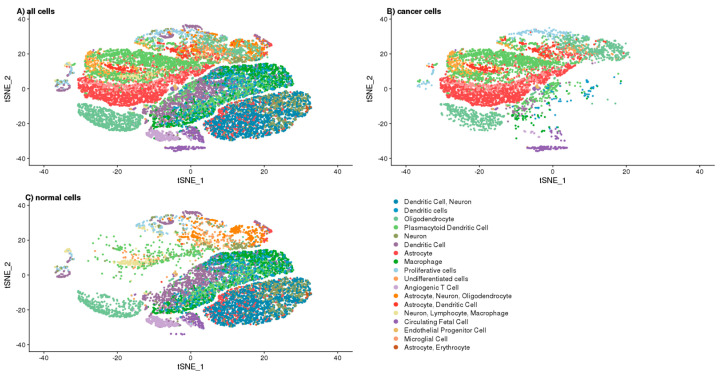
tSNE projection of annotated single cells by expression of markers specific for various cell types. (**A**)—Cancer and normal cells, (**B**)—Cancer cells, (**C**)—Normal cells.

**Figure 6 ijms-22-13217-f006:**
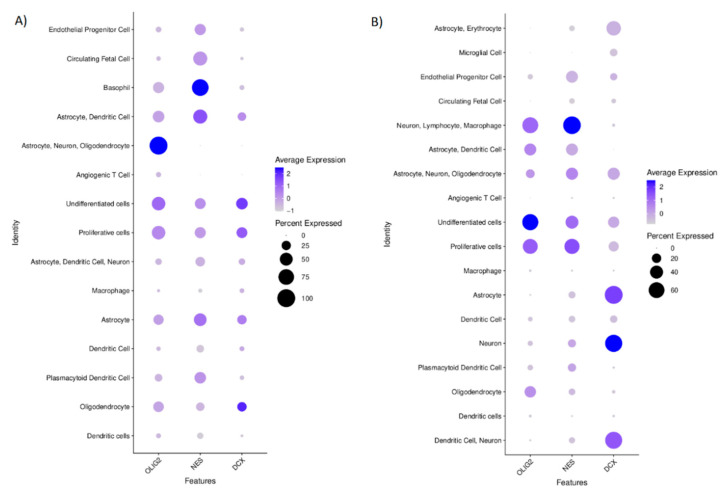
The mean expression of OLIG2, NES, and DCX transcripts among the annotated clusters of single cells that express markers for each cell type. (**A**) the cells annotated as “cancer”, (**B**) the cells annotated as “normal”.

## Data Availability

Single-cell RNA-seq data (EGAD00001006206) can be accessed via European Genome-Phenome Archive. A.O. acquired written permission to use mentioned data.

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
