# Peer review of "Profiling Glioblastoma Cases with an Expression of DCX, OLIG2 and NES"

_ijms, 2021, doi:10.3390/ijms222413217_

Round 1

Reviewer 1 Report

In the present article, the authors describe the role of DCX in Glioblastoma.

I have included some modifications for the authors consideration.

Major comments:

The conclusion is not well documented. First, authors say that DCX is expressed in non-cancerous cells, and then, they conclude that glioma cells express DCX.

In the results sections, authors explained that they use ONLY the canonical aberrations to identify cells as cancerous. I would recommend using another cut-off less restrictive, and compare both results. Perhaps the cells clones that are migrating, or are DCX positives don’t have these aberrations.

Minor comments:

Please unify abbreviations, for example in abstract, you define the abbreviation: “glioblastoma (GBM)” and then, continue using glioblastoma. Please, apply consistently throughout all the manuscript.

You should include in materials and methods, how many microscope images have been analyzed, and how many times were the experiment repeated.

Figures 2 and 3 are not clear, please change the size and remove the grey background.

Figure S3. Please, order the chromosomes.

Author Response

We'd like to thank the Reviewer for his thorough and meticulous review followed by valid comments and suggestions that will certainly improve our manuscript. Below are our detailed responses:

  1. The conclusion is not well documented. First, authors say that DCX is expressed in non-cancerous cells, and then, they conclude that glioma cells express DCX. 
    We pointed out that although most cells were marked as non-cancerous origin, we were able to detect four subpopulations marked as GBM origin (line 226). Therefore, our general conclusion is that (line 272-273), GBM has some cancerous cells with DCX, but it is a small group of cells. It is even more apparent when we consider figure 7. The strongest expression of DCX, with the highest prevalence in ‘normal’ cells, is observed in the cells marked as neurons, which was also suggested by Santra et al. One possibility of ‘normal’ DCX+ cells influx in tumor might be brain trauma caused by expansion of the tumor itself. For example, Itoh et al. in ‘Immature and mature neurons coexist among glial scars after rat traumatic brain injury’ (DOI: 10.1179/016164107X208086) stated that ‘Between 1 and 30 days after injury, doublecortin (DCX)-positive cells were present around the damaged area’. Simultaneously, DCX+ cells with canonical mutations are not so prevalent while being annotated as astrocytes/oligodendrocytes, not neurons.
  2. Please unify abbreviations, for example in abstract, you define the abbreviation: “glioblastoma (GBM)” and then, continue using glioblastoma. Please, apply consistently throughout all the manuscript. 
    Appropriate changes were made in the manuscript.
  3. In the results sections, authors explained that they use ONLY the canonical aberrations to identify cells as cancerous. I would recommend using another cut-off less restrictive, and compare both results. Perhaps the cells clones that are migrating, or are DCX positives don’t have these aberrations.
    We want to thank the reviewer for such a valuable comment. While working on the single-cell data, we considered a different analytical strategy that would abolish the approach used initially by Couturier et al., who first worked on this dataset to annotate cancerous and normal cells in glioma tumors. To distinguish cells as 'normal' or 'cancer' in GBM, we were seeking to use a much less restrictive approach, including non-canonical aberrations (other than a duplication of chr7 or loss of chr10). To this end, we calculated the CNV score as the squared sum of scaled CNV levels for every chromosome, which corresponds to the overall number of aberrations in the cell. Unfortunately, a higher CNV score was not observed in cells showing chr7 duplication or chr10 loss (attached figure).
    Moreover, a higher CNV score was found in one cluster of cells from different tumor samples. Bearing in mind the report of Couturier et al., where a cluster containing cells from other patients was shown to have only the normal cells (Couturier et al., Figure S1b, S1d), we concluded that the overall CNV score does not reflect glioma biology. We agree with the reviewer's comment, and we are aware that tumor cells in GBM may have a CNV pattern that differs from the canonical one, and it is possible to lose some tumor cancer cells. However, the use of a canonical strategy eliminates cells that have been falsely annotated as cancer cells, so we felt this was an appropriate method, especially when the number of patients in the dataset is limited and the study's main aim was not to identify non-canonical CNVs in GBM.
  4. Figures 2 and 3 are not clear, please change the size and remove the grey background 
    Appropriate changes were made in the manuscript.
  5. You should include in materials and methods, how many microscope images have been analyzed, and how many times were the  experiment repeated.
    Appropriate changes were made in the manuscript. (lines 321-322)
    Each of 60 ROIs corresponded to microscope images. From each ROI, up to three 600 px square were taken to perform further analysis. The nature of microscopic slides sometimes prevents marking areas big enough to be sufficient for three squares.
  6. Figure S3. Please, order the chromosomes - a new figure has been submitted with revised manuscript.

Reviewer 2 Report

The authors present a manuscript that in this reviewer’s opinion attempt to classify GBM on the basis of antigen co-staining.     While it is appreciated that this study was performed on human sample, it would be helpful for the authors to clarify the subcategories of the experimental groups.    As it stands the conclusion is that OLIG+DCX+ cells are a marker of tumor-associated reactive astrocytes while nestin+DCX+ Cells are the glioma population of the tumor. Is this correct?   Please better clarify “Tumor-non GBM”   Please clarify is any fresh frozen sample we’re obtained from study   Please indicate a mechanism of action. Are nestin+DCX+ Cell present in human culture cell lines?   In any of the analysis are DCX associated genes aberrantly regulated in the groups noted on figure 1    Can you perform any form of block control to show no artifact staining occurs with the antibodies used on parrafin sections   Are these samples paired from the sample patient?    Please address some of these concerns 

Author Response

Please find response in an attached PDF file

Round 2

Reviewer 2 Report

The reviewer appreciated the revisions made to the manuscript.

The authors are making clear in figure 2 and in the new methods explanation that

OliG2/nestin double positive cells are more associated with normal stroma as compared to OLIGO2-/DCX+ cells (which are still presumably low grade or reactive astrocytes), as compared to nestin /DCX double positive cells expressed only in GBM.

in so far as the descriptive nature of these cells types, the separation on non-tumor GBM from GBM appears dubious and may result in an inappropriate assessment of the cell types present in aggressive / malignant astrocytes (GBM)  as compared to low grade astrocytomas.

Furthermore, the presence of no culture cell line with DCX/NES expression appears counterintuitive as most cells lines on plastic are motile in fashion, and as the authors indicated "DCX was highly expressed in both high-grade and low-grade invasive."

The overall study would be strengthened by assessing more than just 150 random histology sections/spots, focusing on 60 ROIs.  Meaning one may require 200-400 assessments on more than a thousand ROIs.

The other approach more aptly employed would be to develop the histological scoring system of DCX mentioned in the manuscript and then employ this staining strategy on an independent cohort in of tumor samples in a blinded fashion. .  These validation studies are important and expected even in short clinical reports.

Author Response

The authors once again would like to thank the Reviewer for valuable comments. As to related to the Reviewer's comment regarding figure 2 interpretation it's confounding nature pertains to the fact that the presented plot that shows DCX and NES expression levels among the commonly used cell lines was created based on the Protein Atlas database, therefore it shows only a limited number of the cell lines. Thus, some other DCX/NES positive cell lines exist and may be considered as a model for studying the functions of these genes. Additionally, the cell lines do not present tumor heterogeneity, which seems to be their most significant limitation, so it might be challenging to capture these clones expressing both proteins without using the methodology to track single cells. One possibility to omit this would be using the primary cell lines or glioma organoids as a model, thus we are considering those approaches in future studies.

Regarding the methodology of histology sections assessment, we wanted to avoid methods that rely primarily on the investigator's interpretation, such as focusing on overall slide intensity and classifying it into arbitrary brackets. Additionally, although immunofluorescent imaging of tumour sections is a powerful technique, many researchers do not advise it to be used quantitatively, with photobleaching as a primarly factor. For this reason, we developed a technique to count these cells which express DCX instead of basing only on the fluorescence intensity measurements. Still, we are planning to explore in more depth glioblastoma from this and similar aspects. Furthermore, we will enhance our methodology to comply with the reviewer’s suggestions by increasing the number of ROIs and finding an automatic method to capture hundreds or even thousands of random samples of slides. This methodology will subsequently be implemented in a separate validation study suggested by the Reviewer in order to validate it for further use in more robust clinical cohorts.